# In Vitro and In Silico Investigation of Diterpenoid Alkaloids Isolated from *Delphinium chitralense*

**DOI:** 10.3390/molecules27144348

**Published:** 2022-07-07

**Authors:** Shujaat Ahmad, Manzoor Ahmad, Mazen Almehmadi, Syed Adnan Ali Shah, Farman Ali Khan, Nasir Mehmood Khan, Asifullah Khan, Mustafa Halawi, Hanif Ahmad

**Affiliations:** 1Department of Pharmacy, Shaheed Benazir Bhutto University, Sheringal Dir (Upper) 18000, Khyber Pakhtunkhwa, Pakistan; shujaat@sbbu.edu.pk; 2Department of Chemistry, University of Malakand, Chakdara Dir (Lower) 18550, Khyber Pakhtunkhwa, Pakistan; zainabuom2013@gmail.com; 3Department of Clinical Laboratory Sciences, College of Applied Medical Sciences, Taif University, P.O. Box 11099, Taif 21944, Saudi Arabia; dr.mazen.ma@gmail.com; 4Faculty of Pharmacy, Universiti Teknologi MARA Cawangan Selangor Kampus Puncak Alam, Bandar Puncak Alam 42300, Selangor, Malaysia; syedadnan@uitm.edu.my; 5Atta-Ur-Rahman Institute for Natural Products Discovery (AuRIns), Universiti Teknologi MARA Cawangan Selangor Kampus Puncak Alam, Bandar Puncak Alam 42300, Selangor, Malaysia; 6Department of Chemistry, Shaheed Benazir Bhutto University, Sheringal Dir (Upper) 18000, Khyber Pakhtunkhwa, Pakistan; alinuml@gmail.com; 7Department of Agriculture, Shaheed Benazir Bhutto University, Sheringal Dir (Upper) 18000, Khyber Pakhtunkhwa, Pakistan; nasir@sbbu.edu.pk; 8Department of Biochemistry, Abdul Wali Khan University Mardan, Mardan 23200, Khyber Pakhtunkhwa, Pakistan; asif@awkum.edu.pk; 9Department of Medical Laboratory Technology, College of Applied Sciences, Jazan University, Jazan 45142, Saudi Arabia; halawim@jazanu.edu.sa

**Keywords:** diterpenoids, X-ray structure, acetylcholinesterase (AChE), butyrylcholinesterase (BChE) inhibition, *Delphinium*
*chitralense*

## Abstract

This study reports the isolation of three new C_20_ diterpenoid alkaloids, Chitralinine A–C (**1**–**3**) from the aerial parts of *Delphinium chitralense*. Their structures were established on the basis of latest spectral techniques and single crystal X-rays crystallographic studies of chitralinine A described basic skeleton of these compounds. All the isolated Compounds (**1**–**3**) showed strong, competitive type inhibition against acetylcholinesterase (AChE) and butyrylcholinesterase (BChE) in comparison to standard allanzanthane and galanthamine however, chitralinine-C remained the most potent with IC_50_ value of 11.64 ± 0.08 μM against AChE, and 24.31 ± 0.33 μM against BChE, respectively. The molecular docking reflected a binding free energy of −16.400 K Cal-mol^−1^ for chitralinine-C, having strong interactions with active site residues, TYR334, ASP72, SER122, and SER200. The overall findings suggest that these new diterpenoid alkaloids could serve as lead drugs against dementia-related diseases including Alzheimer’s disease.

## 1. Introduction

*Delphinium chitralense* is a high altitude (1520 to 1830 m) annual herb, belonging to the family Ranunculaceae. The roots of *D. denudatum* have been found to possess anticonvulsant properties are commonly utilized in Pakistan in the Unani system of medicine [1]. Phytochemically, the genus *Delphinium* is reported to be a rich source of pharmacologically active diterpenoid alkaloids that pronounce potent antipyretic and analgesic activities. The nor-diterpenoid alkaloids have been found to show ten times more toxic effects as compared to any of the tested alkaloids [2]. A large number of natural products acting as cholinesterase inhibitors, especially diterpenoids and norditerpenoids alkaloids, have been investigated and isolated from different plant species as described extensively in recent studies [3,4,5,6]. In the past decade, a large number of alkaloids possessing C_20_ and C_19_ diterpenoid skeletons (Figure 1) have been isolated from different species of *Delphinium* [7]. Some of delphinium alkaloids are strong inhibitors of cell death, caused by oxidative stress in H9C2 cells [8]. A C-20 diterpenoid alkaloid, deoxylappaconitine, showed very strong analgesic activities, higher than the standard drug, lappaconitine [9]. Moreover, these alkaloids have been under investigation for their cardiac, relaxant, and anti-inflammatory properties, as well as antiproliferative activities against numerous cancer cell lines [7]. Some of the effects have been well documented using the structural activity relationship as it has been observed that the OH groups at positions 1, 8, and 14, as well as N-CH_3_ or N-H are necessary for their various therapeutic cardiac effects [10]. In the early 1990s, methyllyconitine, a major constituent, was found to be an effective ligand for neuronal nicotinic acetylcholine receptor which prompted the scientists to find suitable natural cholinesterase inhibiters to treat cerebral dementia as well as Alzheimer’s diseases (AD) [7].

Alzheimer’s disease (AD) is considered to be one of the most closely related forms of dementia to neurodegeneration disorders. The main cause of AD is the presence of the AChE enzyme which hydrolyzes acetylcholine and is present in at the neuromuscular junction of brain synapses [11]. The abnormal decrease in brain activity with regards to cholinergic function can cause memory defacement in senile dementia disease [12]. In AD, the decline of cognition is connected to the immediate loss of cholinergic neurons and shortage of ACh, which is enhanced by the neuronal ACh. The AChE generally regulates the quantity of ACh in the brain. Consequently, the deficiency of ACh in cells is redressed by the inhibition of the AChE enzyme to improve cognitive abilities. In severe cases of AD, the AChE level is decreased by up to 90% in comparison to a normal healthy brain [12], leading to uncontrollable alterations in the last stages of AD. Most of the literature reports show that a sufficient quantity of BChE is stored in Alzheimer’s plaques in comparison to the quantity of plaques present in normal healthy brains. A number of synthetic compounds such as donepezil, tacrine, and rivastigmine have been applied for the treatment of memory impairment and cognitive dysfunction [13], but these compounds were found to have antagonistic effects, including gastrointestinal complications and problems related to bioavailability [14]. Cholinesterase inhibitors obtained from plants such as jadwarine-A, jadwarine-B, 1β-hydroxy,14β-acetyl condelphine [15], swatinine-C, hohenackerine, aconorine, and lappaconitine [16] are found to be potent and demonstrate competitive and non-competitive enzyme inhibition. The investigation of natural cholinesterase inhibitors is a big task in the area of drug development, particularly for the treatment of Alzheimer’s and other related diseases [17,18].

In the current work, we describe the isolation, structure elucidation, and in vitro and in silico anticholinesterase inhibitory potential of three new C_20_ diterpenoids alkaloids isolated for the first time from *D. chitralense*. The crystal structure determination and DFT calculations of Compound **1** have also been discussed.

## 2. Results and Discussion

### 2.1. Structure Elucidation and Identification

Three new Compounds (**1**–**3**) were isolated by the procedures (see material and methods section) from the aerial parts of *D. chitralense* (Figure 2).

The molecular formula for Compound **1** (C_21_H_27_NO_5_) was established on the basis of its molecular peak in HR-EIMS [M^+^] at *m*/*z* 373.4405 (calcd. 373.4417) and NMR spectral data including single X-ray crystallography.

The NMR spectral data of Compound **1** displayed signals of *N*-methyl at *δ*_H_ 2.25 (3H, s, CH_3_-N); *δ*_C_ 48; methyl group at *δ*_H_ 1.17 (3H, s, H-18); *δ*_C_ 27.3 (C18); a terminal methylene proton at *δ*_H_ 5.07 (H-17a) and 4.80 (H-17b), *δ*_C_ 110.3 (C-17), two oxygenated methines at *δ*_H_ 4.22 (H-1); *δ*_C_ 78.4 (C-1), and *δ*_H_ 4.67 (H-2); *δ*_C_ 75.9 (C-2) along with additional signals of five methylene, four methines, and seven quaternary carbons. These structural features were suggestive of the structure of C_20_ diterpenoid alkaloid. Long-range ^1^H-^13^C correlation of **1** (Figure 3) was obtained through the heteronuclear multiple bond correlations (HMBC) experiment which suggested that H-15 (*δ*_H_ 2.59) signal interacted with C-9 (*δ*_C_ 78.6), C-10 (*δ*_C_ 39.7) and C-1 (*δ*_C_ 78.4), as well as signal of H-5 (*δ*_H_ 2.44) correlated to C-4 (*δ*_C_ 36.7) and C-6 (*δ*_C_ 209.4) (Table 1). The chemical shifts of C-6 and C-13 suggested two ketonic carbonyls in the molecules.

#### 2.1.1. Crystal Structure Determination

Finally, the structure and relative stereochemistry of Compound **1** was established by the study of X-ray diffraction technique (Figure 4).

Compound **1** was crystallized as monoclinic unit of crystal system with *C_2_* space group. The crystal determination and refinement data of isolated natural product (**1**) are tabulated in Table 2 and Appendix A.

Compound **1** contains six main cyclic rings (A–F) (Figure 5). The rings A–E are six-membered rings whereas ring F is five-membered ring. Analyzing the basic/core skeleton, the junction of ring A/E [C-5—C-10—C-20 = 113.4 (4)^0^ ] and B/C [C-8—C-9—C-11 =106.8 (4)^0^ ] are trans fused while rings A/B [C-5—C-10—C-1 =107.2 (4)^0^] and rings E/F [C-5—C-10—C-9 = 110.6 (4)^0^] are cis-fused. The bond lengths and angles were in observed in predicted ranges [19]. The two-hydroxyl group at C-2 and C-9 are *β*-oriented while the other hydroxyl groups at C-1 was noticed to be *α*-oriented. In observing stereochemistry, all the rings showed chair, boat, and half boat conformations. The absolute configuration of Compound **1** cannot be constituted by Mo-Kα diffraction data, although it can be assumed to be equivalent as reported for other isolated C_20_-diterpenoid alkaloids [19]. Keeping in view, the above mentioned spectral and crystal data, the structure of Compound **1** was deduced as 9*β*-hydroxy hetidine (named Chitralinine-A).

#### 2.1.2. DFT Calculation of Compound **1**

The DFT simulations are necessary to gain deeper insights into the molecular structure and electronic properties as it was recently reported in many examples in the literature that DFT simulations were used along with experimental study to obtain the electronic properties which are harder to obtain through experimental analysis. Therefore, we performed the DFT calculations and studied different properties for the studied compound such as HOMO-LUMO analysis, reactivity, global hardness, and optimized structure etc. DFT calculations were generally accomplished on a single unit cell of the molecule [20]. The geometry of Compound **1** with appropriate orientation and spatial arrangement was optimized by using B3LYP-631G (p) and 6-311 + G(d,p)/wB97XD basis sets [21]. The optimized geometry and structure of the compound under investigation is shown in Figure 6. The electronic properties, calculated energy and other relevant parameters are given in Table 3.

HOMO-LUMO energy gaps for Compound **1** were obtained as 0.191 au at 6-31G(d)/B3LYP while 0.292 au at 6-311 + G(d,p)/wB97XD, the values obtained through 6-311 + G(d,p)/wB97XD are higher than 6-31G(d)/B3LYP level because the wB97XD theory has high Hartree–Fock (HF) character and can effectively capture co-relations factors. The values of the E_H-L_ gaps of the studied compound obtained through wB97XD functional with a larger basis set is higher than that of B3LYP functional with a smaller basis set. Because B3LYP functionals overestimated the electronic properties due to lower HF character, this cannot effectively capture electronic co-relation factors. The values of the E_H-L_ gaps studied at wB97XD/6-311 + G(d,p) functional are reported in Table 3.

Moreover, it was observed that the stability of LUMO was mainly due to the electron-accepting properties while the HOMO orbital is usually responsible for the electron-donating ability of inhibitor molecule. Moreover, the greater values of HOMO are signs of electrons donation to the un-occupied orbital of the receptor.

Compound **2** showed specific rotation [α]_D_^30^: −25^0^ (*c* = 1, CHCl_3_). Its molecular formula (C_21_H_27_NO_8)_ was deduced by HR-EIMS (*m*/*z* 421.4550; calcd. 421.4396), higher than chitralinine-A, probably due to the presence of additional hydroxyl groups NMR spectrum of Compound **2** displayed a terminal methylene group at *δ*_H_ 5.02 (H-17a) and 4.85 (H-17b); *δ*_C_ 110.4 (C-17), methyl protons at *δ*_H_ 1.16 (H-18); 29.4 (C-18), four oxygenated methines at *δ*_H_ 4.16 (H-1); *δ*_C_ 75.4 (C-1), *δ*_H_ 4.64 (H-2); *δ*_C_ 73.3 (C-2), *δ*_H_ 3.36 (H-11); *δ*_C_ 80.7 (C-11) and *δ*_H_ 5.0 (H-19); *δ*_C_ 94.2 (C-19), respectively. The ^1^H and ^13^C-NMR data were very similar to those of Compound **1**, suggesting that Compound **2** should also be a diterpenoid and structurally related to Compound **1**. While comparing the chemical shifts of skeletal carbons in Compounds **1** and **2**, the main differences between their ^13^C-NMR data (Table 1) are that there were three more oxygenated functionalities, and additional hydroxyl groups appeared in Compound **2**. To verify further the locations of groups and functionality, HMBC experiment was performed (Figure 3). The hydroxyl group on C-1 was assigned by the HMBC correlations of H-1 (*δ*_H_ 4.16) to C-2 (*δ*_C_ 73.3) and C-10 (*δ*_C_ 43.6). Similarly, other correlations were observed between H-7 (*δ*_H_ 2.05) to C-6 (*δ*_C_ 209.4) and C-8 (*δ*_C_ 36.7); H-14 (*δ*_H_ 2.26) to C-8 (*δ*_C_ 36.7) and C-20 (*δ*_C_ 70.1). On the basis of above physical and spectroscopic data, the structure was deduced as 9,11,12,19*β*- tetrahydoxy hetidine (chitralinine-B).

The molecular formula (C_21_H_29_NO_7_) for Compound **3** was deduced from its HR-EIMS at *m*/*z* 407.4550 (calcd. 407.4563) which was consistent with its ^1^H and ^13^C NMR data (Table 1 and Table 2).

From the NMR spectrum of Compound **3**, the terminal methylene group as existing in Compounds **1** and **2** was inferred on the basis of signals for C-17 methylenic protons singlets at *δ*_H_ 5.02 (H-17a) and 4.85 (H-17b), *δ*_C_ 110.4 (C-17) (Table 1 and Table 2). From the ^1^H-NMR spectrum, two oxymethine protons displayed singlets separately at *δ*_H_ 4.97 (H-19); *δ*_C_ 94.2 (C-19), *δ*_H_ 3.27 (H-11); *δ*_C_ 80.7 (C-11), confirmed the presence of hydroxyl groups in Compound **3** at C-19 and C-11. The ^13^C-NMR spectrum of Compound **3** showed twenty-one signals for primary, secondary, tertiary and, quaternary carbon atoms. Compared to the chemical shifts of skeletal carbons in Compounds **2** and **3**, the main difference between their ^13^C NMR data (Table 1) is that Compound **3** lacked ketonic functionality at C-13. After describing the skeleton of Compound **3**, long range HMBC interaction (Figure 3) permitted the placement of the hydroxyl substituent at their respective positions, as the H-1 (*δ*4.19) shows ^2^*J* correlations with C-2 (*δ* 73.3) and C-10 (*δ* 44.0). Similarly, the HMBC coupling of terminal methylinic protons (*δ* 5.02 & 4.85) exhibited *^1^J* interaction with C-16 (*δ* 142) and ^*2*^*J* interaction with C-12 (*δ* 78.3) and C-15(*δ* 26.9). All of the above spectral evidence led to the establishment of C_20_ diterpenoid structure of **3** as 1*α*, 2, 9,11,12,19*β*-hexahydoxy atisinone (Chitralinine-C).

### 2.2. Acetylcholinesterase and Butyrylcholinesterase Inhibition Activities

All the natural products isolated from *D. chitralense*, were tested for their enzyme inhibition activity against AChE and BChE, respectively, and showed promising inhibitory potential against both the tested enzymes in vitro. Therefore, it might be concluded that the compounds isolated from *D. chitralense* could be optimized as lead candidates in AD and related ailments. The Compounds (**1**–**3**) were found to be potent against AChE and BchE as compared to standard drugs, showing competitive types of inhibition. The IC_50_ values of Compounds **1**–**3** against AChE were 13.86 ± 0.35, 11.64 ± 0.08 μM and 12.11 ± 0.82 μM while against BChE the values were 28.17 ± 0.92 μM, 24.31 ± 0.33 μM and 26.35 ± 0.06 μM, respectively (Table 4).

These significant results highlighted the interest in isolation and reputation of this class of secondary metabolites present in *D. chitralense*.

### 2.3. Molecular Docking Study

The ligand base docking result of Compound **1** against the acetylcholinesterase target showed binding free energy of −14.457 Kcal/mol (Table 5). The visual inspection of docked compound revealed that it interacts with four residues, i.e., SER200, GLY119, GLY118, and SER122 of acetylcholinesterase (Figure 7).

Details regarding chemical activity of Compound **1** were determined based on the correlation between calculated energies and quantum parameters. Quantum parameters, i.e., electronegativity (χ), electrophilicity (ω), hardness (η), and softness (S), are universal descriptors which are used to explain the chemical behavior of the molecules [22]. The hardness (η) value decides the resistance of an atom for the charge transfer to another atom. The electron-receiving ability of an atom is determined by the softness value. Electronegativity χ is the ability of molecules to attract electrons. The electrophilicity index ω is related to the electrophilic property of a molecule.

The significant orbitals found in molecules that affect the biological activity, molecular reactivity, and other electronic properties are HOMO and LUMO [23,24,25]. Deep insight into the biological mechanism of the active molecules can also be deduced on the basis of frontier orbital energy studies. Figure 5 shows that both HOMO and LUMO are localized at the tertiary nitrogen atom and hydroxyl group of Compound **1**. This makes it clear that the activity related to this molecule could be attributed to the hydroxyl and tertiary nitrogen. Thus, HOMO and LUMO orbitals that penetrate the hydroxyl group will form interactions with active sites of the enzyme, giving rise to its reported biological activity.

The docking result of Compound **2** showed binding free energy of −15.591 Kcal/mol and selective interactions with TYR121, SER122, HIS440, and SER200 residues (Figure 8). Likewise, the docking result of Compound **3** showed binding free energy of −16.400 Kcal/mol and interactions with TYR334, ASP72, SER122, and SER200 (Figure 9). All the Compounds **1**–**3** were found to develop interactions with the key residues of the gorge site of the acetylcholinesterase target and the in silico results are congruent with experimental findings, suggesting the competitive nature of Compounds **1**–**3**. The differences in protein-ligand interaction and binding energies among these compounds are due to differences in basic chemical structures and receptor binding affinities.

## 3. Materials and Methods

### 3.1. General Procedures

The optical rotations ([α]^25^_D_) were obtained through a “JASCO DIP 360 polarimeter (Tokyo, Japan)” while the melting points (mps) were measured using “BioCote Stauart SMP10 (Tokyo, Japan)” melting point instrument. The mass spectral assignments were made from EI-MS/HR-EIMS spectra obtained through “JEOL JMS HX 110 (Tokyo, Japan)” while ^1^H-NMR/^13^C-NMR spectral measurements were carried out by using “Bruker NMR, Germany (500, 600 MHz for ^1^H-NMR; 125, 150 MHz for ^13^C-NMR (δ, ppm)), respectively. FT-IR analyses were determined on “JASCO-320-A spectrophotometer in KBr” as well as “Perkin–Elmer spectrophotometer”. All the solvents used in extraction and isolation of compounds were distilled before use, while the deuterated solvents were used for NMR analysis. Thin layer chromatography (TLC) was carried out using “silica gel F_254_ pre-coated aluminum sheets”. Visualization of TLC was conducted through a UV lamp at both 254 & 366 nm (λ_max_) as well as “Dragendorff’s reagent”. The solvent system; 20% acetone-hexane: 10 drops of diethylamine was used as developing solvent for TLC.

### 3.2. Plant Material

The aerial parts of *D. chitralense* were collected from Kumrat Valley (Latitude = 35.560654; Longitude = 72.200846; Altitude is 8100 feet) of Dir (U). A voucher specimen with number H.UOM. BG-161 was deposited in the herbarium of University of Malakand, Dir (L).

### 3.3. Extraction and Isolation

Standard procedure was adopted in the extraction and isolation processes with some modifications [15]. The shade dried powdered material of (10 kg) of *D. chitralense* was extracted for seven days with 80% methanol; thrice (3 × 20 L) in closed glass containers. This combined methanolic extract was filtered and concentrated in a vacuum on a rotary evaporator (Buchi, Flawil, Switzerland) at 40 °C to obtain 890 g methaloic crude. This methaloic crude was first pooled with 5% HCl solution (pH = 1–2) and then extracted with CHCl_3_ to separate the non-alkaloidal portion from acid-aqueous alkaloidal solution. This acidic portion was basified with 5% NaOH (pH= 8–10) to obtain free alkaloids in the solution. This solution was again re-extracted with CHCl_3_ to obtain alkaloidal portion (18 g). This alkaloidal portion was fractionated though a silica gel (360 g) column, which was eluted with increasing polarities of n-hexane (100%, DX-1) and n-hexane-chloroform and chloroform-methanol (up to 20% methanol) in gradient manner that afforded eight sub-fractions (DX-1 to DX-8). The sub-fraction DX-4 obtained from solvent system n-hexane-chloroform (50:50) showed interesting spots on TLC and was re-chromatographed on the flesh silica gel column eluted with n-hexane:acetone (80:20) with 10 drops of diethylamine (DEA/100 mL) to produce Compound **1** (top fraction) and Compound **2** (tail fraction). The fraction DX-5 which was obtained from major column with n-hexane-chloroform (60:40) yielded a semi pure compound, which was subjected to a flash column using gradient solvent system of n-hexane-acetone-10 drops of DEA. Elution with n-hexane-acetone (80:20) resulted in the isolation of pure Compound **3**.

### 3.4. Physical and Spectroscopic Data of New Compounds

#### 3.4.1. Chitralinine-A (1)

White crystal; m.p: 224–227 ^°^C; [α]_D_^30^: −35^0^ (c = 1, CHCl_3_); IR (υ_max_ cm^−1^): 1720 & 1650 (C = O), 988 (C = CH_2_), 3460, 3422 (OH); ^1^HNMR (600 MHz, CDCl3): δ 5.07, 4.80 (2H, s, -CH_2_), δ 4.67, (1H, t, *J* = 4.7Hz, H-2), δ 4.22 (1H, brd s, H-1), δ 3.59 (1H, br s, H-20), δ 3.04 (2H, s, H-19), δ 2.66, (1H, t, *J* = 11 Hz, H-12), δ 2.59 (2H, d, *J* = 4 Hz, H-15), δ 2.44 (1H, s, H-5), δ 2.31 (2H, t, *J* = 2.1 Hz, H-11), δ 2.25 (3H, s, -NCH_3_), δ 2.19 (1H, d, *J* = 3 Hz, H-14), δ 2.08 (2H, s, H-7), δ 1.82 (2H, m, H-3), δ 1.17 (3H, s, H-18), HR-EIMS *m*/*z*: 373.4405 (C_21_H_27_NO_5_, calcd. 373.4417); ^13^C-NMR (150 MHz/CDCl_3_): See Table 1.

#### 3.4.2. Chitralinine-B (2)

White powder; m.p: 252–255 ^°^C; [α]_D_^30^: −25^0^ (c = 1, CHCl_3_); IR υ_max_ cm^−1^: 1660, 1450 (C = O), 3480, 3360, 3280 (OH) 1030 (C = CH_2_); ^1^HNMR (600 MHz, CDCl3): *δ* 5.02, 4.85 (2H, s-CH_2_), *δ* 5.0 (1H, s, H-19), *δ* 4.64 (1H, t, *J* = 4.9 Hz, H-2), *δ* 4.16 (1H, brd s, H-1), *δ* 3.59 (1H, br s, H-20), *δ* 3.36 (2H, s, H-11), *δ* 2.84 (1H, s, H-5), *δ* 2.46 (3H, s, -NCH_3_), *δ* 2.26 (1H, d, *J* = 2.5 Hz, H-14), *δ* 2.05 (2H, s, H-7), *δ* 1.92 (2H, m, H-3), *δ* 1.89 (2H, s, H-15), *δ* 1.16 (3H, s, H-18) HR-EIMS *m*/*z*: 421.4380 (C_21_H_27_NO_8_, calcd. 421.4396); ^13^C-NMR (150 MHz/CDCl_3_): See Table 1.

#### 3.4.3. Chitralinine-C (3)

Amorphous powder; m.p: 233–238 °C; [α]_D_^30^: −23^0^ (c = 1, CHCl_3_); IR υ_max_ cm^−1^: 1658, 1446 (C = O), 3500, 3350 (br OH), 1032 (C = CH_2_^1^HNMR (500 MHz, CDCl3): *δ* 5.02, 4.85 (2H, s-CH_2_), *δ* 4.97 (1H, s, H-19), *δ* 4.64 (1H, t, *J* = 4.8 Hz, H-2), *δ* 4.19 (1H, brd s, H-1), *δ* 3.57 (1H, br s, H-20), *δ* 3.27 (2H, s, H-11), *δ* 2.45 (3H, s, -NCH_3_), *δ* 2.30 (2H, s, H-7), *δ* 2.24 (1H, s, H-5), *δ* 1.95, 1.93 (2H, dd, 2H), *δ* 1.91 (2H, m, H-3). *δ* 1.89 (2H, s, H-15), *δ* 1.63 (1H, m, H-14), *δ* 1.15 (3H, s, H-18); HREI-MS *m*/*z***:** 407.4550 (C_21_H_29_NO_7_, calcd. 407.4563); ^13^C (125 MHz/CDCl_3_): See Table 1.

### 3.5. X-ray Crystallography

The crystal structure data for Compound **1** was obtained from Single crystal X-ray crystallographic analysis using “STOE-IPDS II (Darmstadt, Germany); Graphite-monochromator at room temperature and Mo-Kα radiation (λ = 0.71073 Å)”. Data were captured by using charge-coupled device (CCD) area detector. The structure was solved and refined though SIR97 [26], SHELXL97 [27] and WinGX [28] programs.

### 3.6. Density Functional Theory (DFT) Calculations

The crystallographic data were used in the DFT calculations of **1** by means of two models of theory; “B3LYP-631G (p) and 6-311 + G(d,p)/wB97XD” [20,26,29]. Furthermore, the same methods were applied in obtaining other related information including HOMO-LUMO energy gap, optimized geometries, electron affinity, electrophilicity, ionization potential, and global hardness [30]. The data were manipulated through “Gauss-view molecule visualizer and GAUSSIAN-03 programs (Wallingford, CT, USA)”. Recently, this procedure has also been applied for DFT calculations and obtaining data of other parameters in C_19_ and C_20_ alkaloids [30].

### 3.7. Acetylcholinesterase (AChE) and Butyrylcholinesterase (BChE) Inhibition Assays

All the solvents used in this assays were of analytical grade while the chemicals/reagents such as AChE (Electric-eel EC 3.1.1.7), BChE (horse serum EC 3.1.1.8), DTNB, Acetyl choline iodide (AChI), butyryl choline chloride (BChI) and the reference, galantamine were purchased from Sigma–Aldrich (St. Louis, MO, USA). The inhibition was obtained through spectroscopic measurements [31]. Standard procedures and conditions of the assays were applied throughout the experiments [32]. Various dilutions of the tested compounds (62.5, 125, 250, 500 and 1000 μg/mL) were used in this assay. AChI and BChl were used as substates in this assay.

In brief, 880 µL of sodium phosphate buffer solution (62 mM, pH 8) containing 0.2 mM DNB was mixed with 40 µL solution of compound and 40 µL AChE or BChE solutions. This reaction mixture was incubated at 25 °C for 15 min followed by initiation of the reaction through addition of acetyl choline (ACh) or butyryl choline (BCh) (40 μL) in each experiment. The formation of yellow colored product (5-thio-2-nitorbenzoate anion) from reaction of DTNB with acetylcholine/butyrylcholine were observed through naked eyes, and the absorbance was measured at 412 nm using UV/Visible spectrophotometer (BMS-USA, New York, NY, USA). All the experiments were carried out in triplicate.

#### IC50 Values Determination

Various dilutions of the tested compounds (62.5, 125, 250, 500, and 1000 μg/mL) were used in this assay for IC_50_ determination. The data (activity in percent vs concentration) were fitted into a non-linear sigmoid plot using MS Excel (Microsoft, Redmond, WA, USA) program. The non-linear concentration-dependent inhibitory concentration of compounds were taken into account at low and high concentration leading to determination of IC_50_ values. The effective concentrations of Compounds **1**–**3** were represented in μM [33].

### 3.8. Molecular Docking Study

To perform molecular docking analysis, the choline esterase crystal structures (PDB ID: 1ACJ) was retrieved from the Protein Databank (PDB). The molecular docking studies were carried out in the presence of water molecules as they play crucial role in the enzymatic activities [34,35,36,37] while other ions were removed from the retrieved crystal structures using the Molecular Operating Environment (MOE) software (www.chemcomp.com, accessed on 10 May 2022). Hydrogen atoms were added to the protein structures by 3D protonation and then energy minimization was carried out using the default parameters of the MOE. The structures of the Compounds **1**–**3** were built and their energy minimization was performed using the default parameters of the MOE. The choline esterase target was allowed to dock to the Compounds **1**–**3** by the default parameters i.e., Placement: Triangle Matcher, Rescoring: London dG of MOE software. The binding pockets are identified by site-finder module of the MOE. For each ligand ten conformations were generated. The top-ranked conformation of each compound was used for subsequent analysis [38].

## 4. Conclusions

In the present research work, bioactivity-guided isolation, crystal structure determination, DFT calculation, anticholinesterase inhibitory potential, and molecular docking studies of diterpenoid alkaloids isolated from *D. chitralense* have been carried out. The structures of three new diterpenoids were established by spectral interpretation, including single X-ray crystallography. Some structural parameters of Compound **1** were calculated by means of DFT. All the isolated compounds were screened for their possible anticholinesterase inhibitory potential and were found to exhibit strong, competitive types of inhibition against cholinesterase as compared to the standard. Further in silico evaluation of isolated compounds exhibited possible binding modes and justified the experimental results. In addition, the negative binding energies of the isolated Compounds **1**–**3** showed proper relationships to the AChE and BChE enzymes. Thus, the present study validates a potential role of diterpenoid alkaloids from *D. chitralense* in the treatment of neurodegenerative disorders and suggests that they could be good natural candidates against AD.

## Figures and Tables

**Figure 1 molecules-27-04348-f001:**
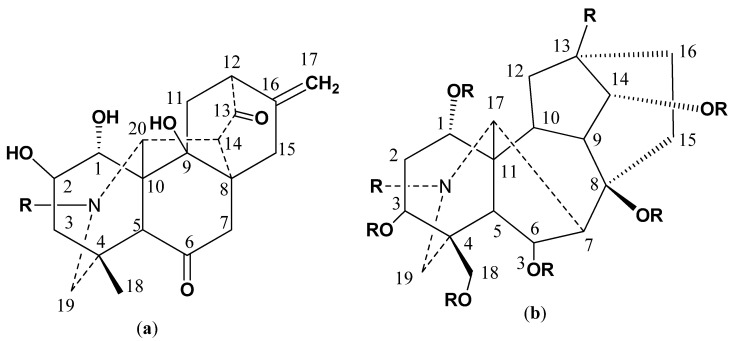
General structures of (**a**) C-20 diterpenoid alkaloids (**b**) C-19 diterpenoid alkaloids.

**Figure 2 molecules-27-04348-f002:**
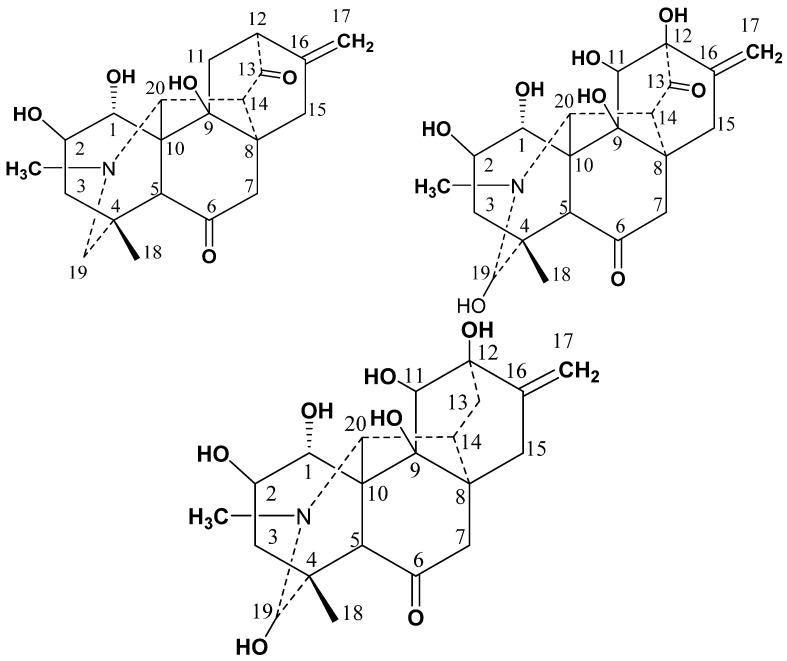
Structures of Compounds **1**–**3**.

**Figure 3 molecules-27-04348-f003:**
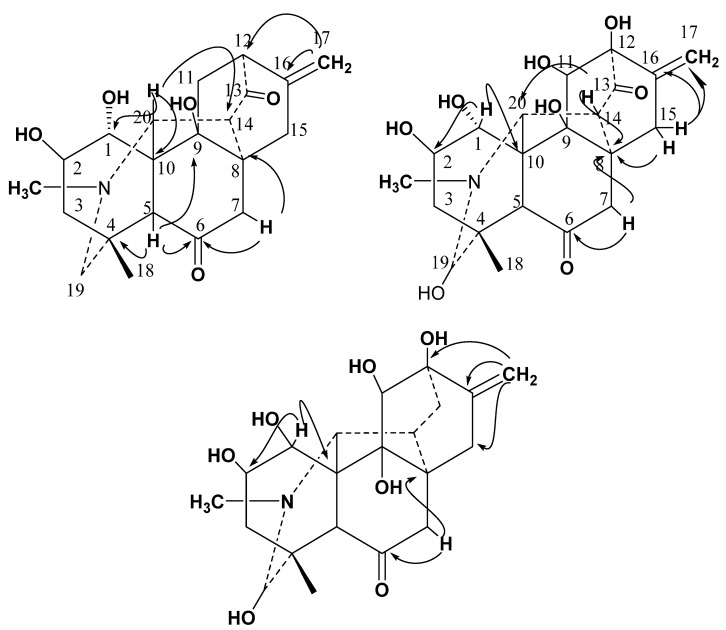
HMBC interaction in **1**–**3**.

**Figure 4 molecules-27-04348-f004:**
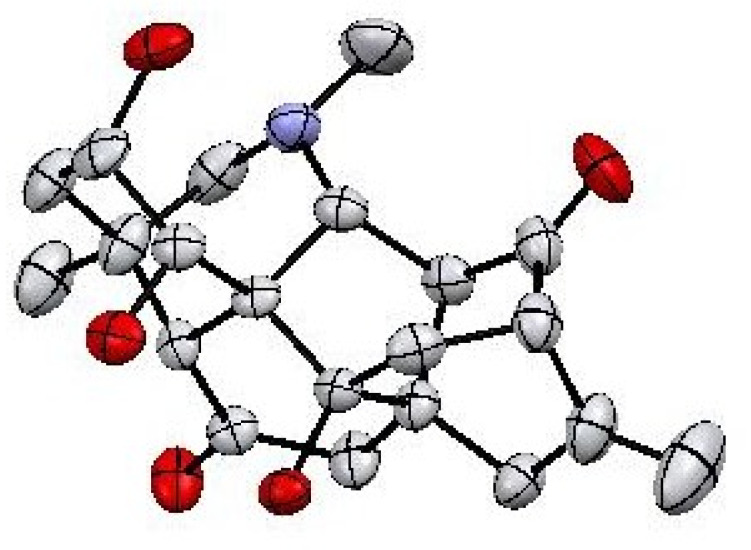
Structural representation of Compound **1**, with 50% probability of thermal ellipsoids, hydrogens were emitted for clarity.

**Figure 5 molecules-27-04348-f005:**
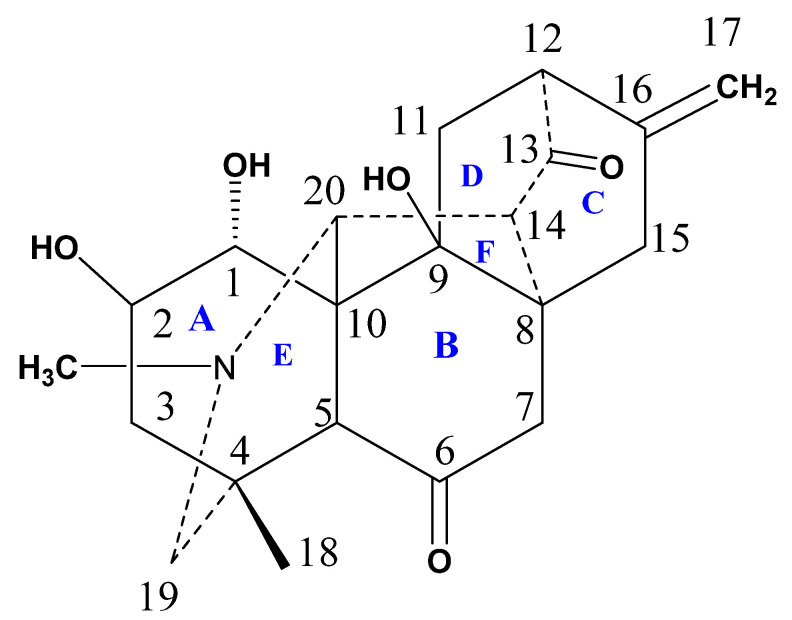
Structure of Compound **1**, orientation of different rings and groups are shown accordingly.

**Figure 6 molecules-27-04348-f006:**
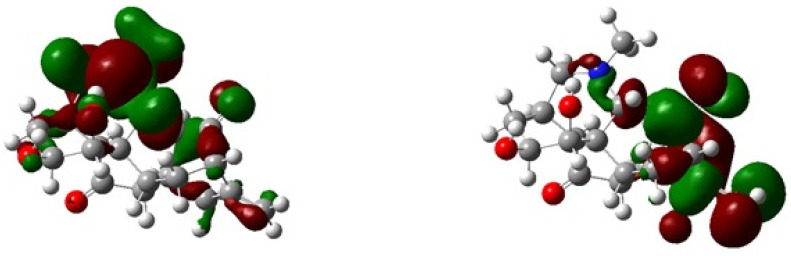
HOMO-LUMO of Compound **1** calculated at B3LYP/6-31þG (p).

**Figure 7 molecules-27-04348-f007:**
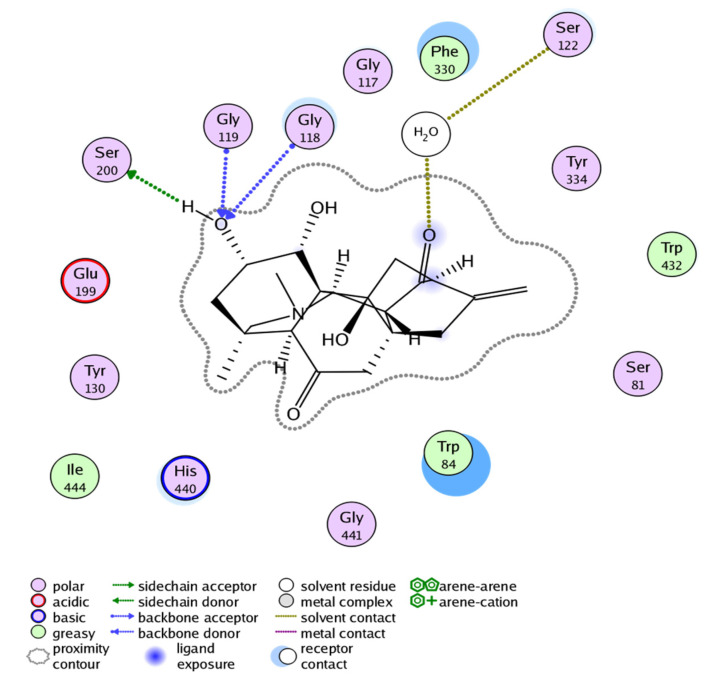
Docking pose of Compound **1** within cholinesterase target. (Docking score of = −14.457).

**Figure 8 molecules-27-04348-f008:**
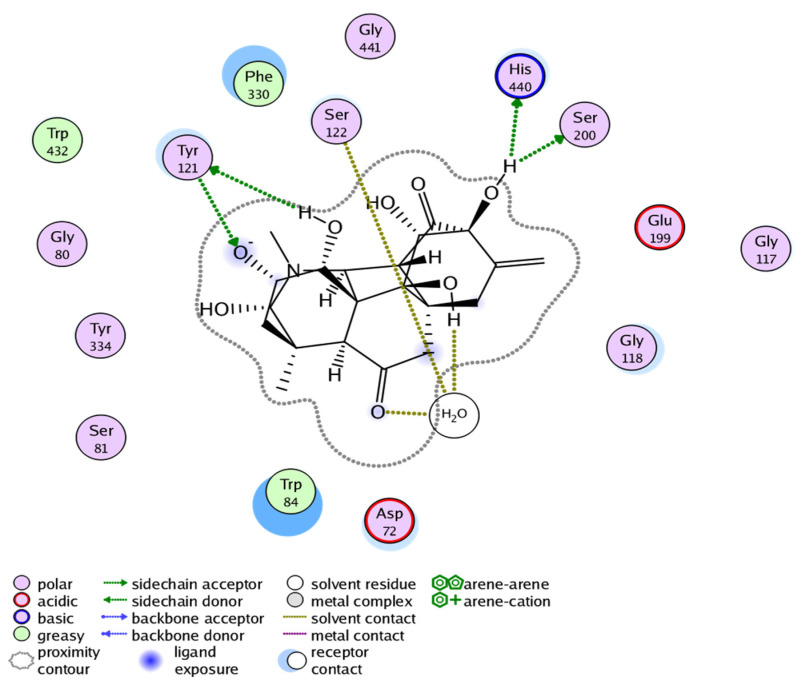
Docking pose of Compound **2** within cholinesterase target (Docking score of = −15.591).

**Figure 9 molecules-27-04348-f009:**
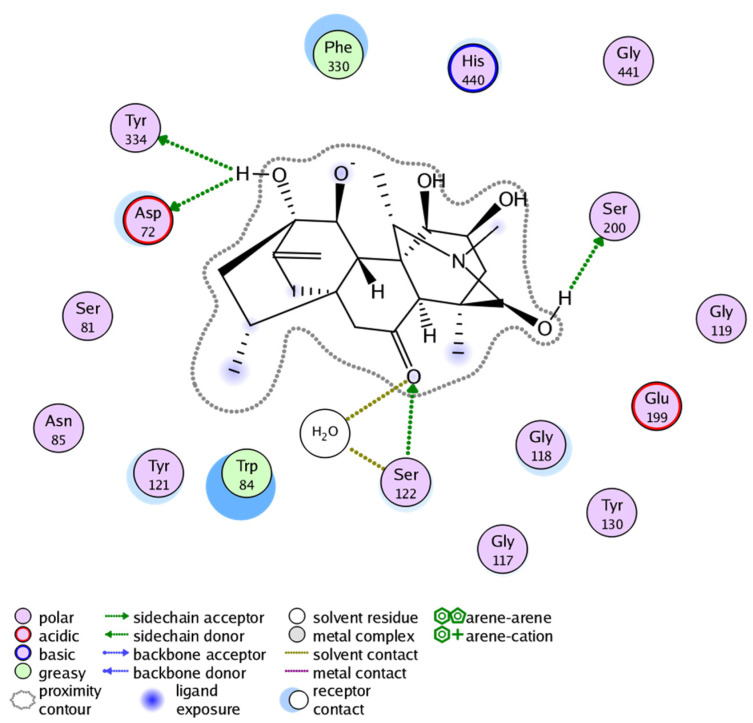
Docking pose of Compound **3** within cholinesterase target (Docking score of = −16.400).

**Table 1 molecules-27-04348-t001:** ^13^C NMR data of Compounds **1**–**3** in CDCl_3_.

Position	Compound 1 (150 MHz)	Compound 2 (150 MHz)	Compound 3 (125 MHz)
1	78.4	75.4	75.7
2	75.9	73.3	73.3
3	40.7	34.0	34.0
4	36.7	32.5	32.6
5	59.9	49.7	48.9
6	209.4	209.4	209.6
7	49.7	44.0	44.2
8	44	36.7	36.7
9	78.6	78.3	83.8
10	39.7	43.6	44.0
11	32.3	80.7	80.7
12	53.4	89.3	78.3
13	209.4	209.5	34.8
14	57.9	48.7	40.7
15	34.9	30.8	26.9
16	142	141.9	142
17	110.3	110.4	110.4
18	27.3	29.4	29.7
19	63.1	94.2	94.2
20	70.3	70.1	70.1
21	48	38.7	38.7

**Table 2 molecules-27-04348-t002:** Crystal data and structure refinement of Compound **1**.

Crystal Parameter Compound 1			
Empirical formula	C_21_H_27_NO_5_	Density (mg m^−3^)	1.153
Formula weight	373.43	(h, k, l) min	(−31, −5, −15)
Temperature (K)	29.6	(h, k, l) max	(31, 9, 15)
Wavelength (Å)	0.71073	Theta (max)	26.0
Crystal system	Monoclinic	R (reflection)	0.053(2408)
Space group	*C2*	wR_2_	0.185
A	25.726 (5) Å	No of measured, independent and observed [*I* > 2σ(*I*)] reflections	8474, 3521, 2408
B	7.5766 (12) Å	*R_int_*	0.053
C	12.654 (2) Å	(sin θ/λ)_max_ (Å^−1^)	0.617
Volume Å^3^	2150.4 (6) Å^3^	No. of reflections	3521
μ (mm^−1^)	0.08	No. of parameters	270
Z	4	No. of restraints	1
Crystal size (mm)	0.43 × 0.22 × 0.18	Absolute structure parameter	−0.7 (10)

**Table 3 molecules-27-04348-t003:** Calculated chemical parameters of Compound **1** computed at various level of DFT and basis sets, with values shown in atomic unit (au).

Compound-1	6-31G(d)/B3LYP	6-311 + G(d,p)/wB97XD
E_HOMO_ (au)	−0.225	−0.319
E_LUMO_ (au)	−0.034	−0.027
ΔE = (E_LUMO-_E_HOMO_) (au)	0.191	0.292
IE = = −E_HUMO_ (au)	0.225	0.319
EA = −E_LUMO_ (au)	0.034	0.027
Global Hardness(η) = 1/2 (E_LOMO_-E_HOMO_)	0.095	0.146
Chemical Potential μ = 1/2 (E_HOMO_ + E_LUMO_)	−0.095	−0.146
Global Electrophilicity ω = μ^2^/2η	0.048	0.073

**Table 4 molecules-27-04348-t004:** AChE and BChE inhibitory activities of alkaloids from *D. Chitralense*.

S. No	Compounds	AChE ± SEM ^a^ (μM)	BChE ± SEM ^a^ (μM)	Type of Inhibition
1	Chitralinine A	13.86 ± 0.35	28.17 ± 0.92	Competitive
2	Chitralinine B	11.64 ± 0.08	24.31± 0.33	Competitive
3	Chitralinine-C	12.11 ± 0.82	26.35± 0.06	Competitive
6	Allanzanthane A	8.23 ± 0.01	18 ± 0.06	
7	Galanthamine ^b^	10.12 ±0.06	20.62 ± 0.08	

^a^ Standard error of mean of five assays; ^b^ Positive control used in the assays.

**Table 5 molecules-27-04348-t005:** Interaction features of Compounds **1**–**3** against Cholinesterase.

Inhibitors	MOE Score	MOE Interactions Residues	Gorge Site Residues of Target
1	−14.457	Ser200, Gly119, Gly118, Ser122	121(288)297(290)120(118)121(119)204(201)447(440)334(327)203(200)86(84)72(70)124(121)286(279)
2	−15.591	Tyr121, Ser122, His440, Ser200	
3	−16.400	Tyr334, Asp72, Ser122, Ser200	

## Data Availability

All data contained within this article.

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
