# Peer review of "In Vitro and In Silico Investigation of Diterpenoid Alkaloids Isolated from Delphinium chitralense"

_molecules, 2022, doi:10.3390/molecules27144348_

Round 1

Reviewer 1 Report

The over all finding suggests that these new diterpenoid alkaloids could serve as lead drugs against dementia related diseases including Alzheimer’s disease....Quite a big statement as per the contents reported in this manuscript.

The role of related molecules to Alzheimer's  disease is lacking in the introduction part. Needs to be rewritten with the  proper citations.

Isolation & Extraction methodology part is not clear, better to rephrase the paragraph or may be use schematic  diagram for better representation. Please refrain using chloroform  in a laboratory setting, rather it may replaced with dichloromethane.  How do you eliminate DEA from the fractions still not mentioned.

Strengthen your discussion part with prior the  research. 

Conclusion part can be rephrased according to title and objectives

Author Response

Response to reviewer 1 comments/suggestions:

Thanks for providing us the opportunity to revise our manuscript entitled “In vitro and In Silico Investigation of Diterpenoid Alkaloids Isolated from Delphinium chitralense ” We agree with the comments and concerns as pointed by your good self. We have addressed them in earnest in this revision. The corrections and suggestions were made in yellow highlight in the revised article and the responses to each comment are addressed below. Please let us know if any other things are necessary for publication. Thank you for all the efforts spent on our manuscript. We are looking forward to your decision on our revised manuscript.

Main Comment:

The overall finding suggests that these new diterpenoid alkaloids could serve as lead drugs against dementia related diseases including Alzheimer’s disease..Quite a big statement as per the contents reported in this manuscript.

Reviewer Comment: The role of related molecules to Alzheimer’s disease is lacking in the introduction part. Needs to be rewritten with the proper citations.

Authors response: Introduction is revised as per suggestion.

Reviewer Comment: Isolation & Extraction methodology part is not clear, better to rephrase the paragraph or may be use schematic diagram for better representation. Please refrain using chloroform in a laboratory setting, rather it may be replaced with dichloromethane.  How do you eliminate DEA from the fractions still not mentioned?

Authors response: The isolation and extraction procedures have been revised and modified according to the suggestions. DEA, have often been applied to facilitate the separation of alkaloids in TLC as well as in column. It was evaporated during the concentration process of our compounds along with eluting solvents.

Reviewer Comment: Strengthen your discussion part with prior the research. 

Authors response: Both results and discussions are merged into single heading “Results and discussion” as per reviewer 3 comment. Moreover, the discussion has been amended according to the suggestions.

Reviewer Comment: Conclusion part can be rephrased according to title and objectives 

Authors response: Conclusion rephrased as per reviewer suggestion.

Reviewer 2 Report

The authors present a manuscript entitled “In vitro and in silico investigation of diterpenoid alkaloids isolated from Delphinium chitralense”. Three molecules diterpene alkaloids (chitralinine A-C, 1-3) have been isolated and determine the structure using IR, NMR, MS, [α]D, and X-ray. Some suggestions are required to improved the quality of manuscript:

1.            Standard format for reporting original research is required to improve. A lot of mistakes are found in the manuscript, for example: the order of table (Tables 5 and 6), the expression of NMR Table 1 (position, δC, mult., δH J in Hz). It is not necessary to explain the NMR field in the table. It is enough to explain in Material and Method. Expression of Table for crystallography. Please use a simple Table to show the result. It is not necessary to use box for Figure 2. The use of capital in Table 6 should be changed. The quality of figures needs to improve. Please use arrow correctly when authors explain HMBC (Figure 2). In addition, many grammatical errors are found.

2.     Authors requires to draw the chemical structures uniformly (size of atom, stereochemistry descriptors, etc). Moreover, authors require to give some atomic number on the chemical structures especially in Figure 1 and 2. This will help to understand the explanation of HMBC correlations.

3.      The DFT calculation should be improved using triple zeta basis with dispersion correction.   

After significant improvement, this manuscript can be considered to next step.  

Author Response

Response to reviewer 2 comments/suggestions:

Thanks for providing us the opportunity to revise our manuscript entitled “In vitro and In Silico Investigation of Diterpenoid Alkaloids Isolated from Delphinium chitralense ” We agree with the comments and concerns as pointed by your good self. We have addressed them in earnest in this revision. The corrections and suggestions were made in yellow highlight in the revised article and the responses to each comment are addressed below. Please let us know if any other things are necessary for publication. Thank you for all the efforts spent on our manuscript. We are looking forward to your decision on our revised manuscript.

Main Comment:

The authors present a manuscript entitled “In vitro and in silico investigation of diterpenoid alkaloids isolated from Delphinium chitralense”. Three molecules diterpene alkaloids (chitralinine A-C, 1-3) have been isolated and determine the structure using IR, NMR, MS, [α]D, and X-ray. Some suggestions are required to improved the quality of manuscript:

Reviewer Comment: Standard format for reporting original research is required to improve. A lot of mistakes are found in the manuscript, for example: the order of table (Tables 5 and 6), the expression of NMR Table 1 (position, δC, mult., δH J in Hz). It is not necessary to explain the NMR field in the table. It is enough to explain in Material and Method.

Expression of Table for crystallography. Please use a simple Table to show the result. It is not necessary to use box for Figure 2. The use of capital in Table 6 should be changed. The quality of figures needs to improve. Please use arrow correctly when authors explain HMBC (Figure 2). In addition, many grammatical errors are found.

Authors response: The manuscript has been revised according to the standard format of journal guidelines. The manuscript is thoroughly checked for all kinds of errors pointed by the reviewer and hence corrected accordingly. Tables 5 and 6 are placed at their respective positions. Spectral data for all compounds in table 1 are shifted to experimental section. Table 3 describing crystal data of compound 1 is revised as per reviewer comment. Boxes are removed from figures as per suggestion. Table 6 revised. Figures quality improved.  Figure 2 may be read as Figure 3 and is revised. Grammatical errors removed.

          Reviewer Comment: Authors requires to draw the chemical structures uniformly (size of atom, stereochemistry descriptors, etc). Moreover, authors require to give some atomic number on the chemical structures especially in Figure 1 and 2. This will help to understand the explanation of HMBC correlations.

Authors response: Figures 1 and 2 may be read as Figure 2 and figure 3 are revised as per your suggestion.

  1. The DFT calculation should be improved using triple zeta basis with dispersion correction.   

The simulations for the studied compound are re-performed using the larger basis set 6-311+G(d,p) and for dispersion correction the wB97XD method is used. As the triple zeta aug-ccpVTZ basis calculations are really expensive and we have limited computer facilities therefore we used 6-311+G(d,P) basis set which have greater accuracy for organic single molecules as reported in lot of literature. The results discussed in the results and discussion section highlighted in yellow.

Reviewer 3 Report

Introduction:

A figure showing the generalised structure of discussed alkaloids should be added, otherwise it is difficult to follow the text, where importance of several functional groups is highlighted.

line 34 - interpunction
lines 36-37 - these details about folk medicine seem to be unnecessary and should be removed

Results:

Figure 1 - no numeration of C atoms, it need to be added in accordance with the numeration used for NMR analysis in tables 1-2.
Figure 2 - no numeration of C atoms...
Figure 3 - very low resolution, it need to be reuploaded in higher quality
Table 3 - a significant number of crystallographic parameters: numbers of reflections, independent reflections, data/restrains/parameters or goodness of fit among others is lacking and need to be provided.
Table 6 - bad formatting, numbering of tables 5 and 6 is wrong, need to be inverted

Following tables (X-ray crystallography of comp. 1) must be provided:
bond lengths, valenca angles, torsion angles, geometry of hydrogen bonds (if applies)

Discussion:

Most of this part need to be moved to "Results" since it is merely description of experimental and computational studies, perhaps both parts can be merged into one - results and discussion. Lack of numbering of carbon atoms on figures makes discussion of NMR spectra very difficult to follow.

Preparation of compounds 1-3 should be moved to Materials and Methods

Description of crystal structure of compound 1 (including its stereochemistry) is very limited. Is there any particular explanation why stereochemistry of 1 cannot be elucidated from X-ray data?

What was the rationale for DFT calculations? The idea behind this part of the study and the results are almost neither presented nor discussed in the text

On which base the mechanism of inhibition (competitive for compounds 1-3) was inferred. Kinetic plots for enzymatic assays should be also provided here or in the supplementary.

What is the unit of energy obtained for docking? Why there is almost no discussion concerning the mechanism of binding and differences between protein-ligand interactions for each compound.

Materials and methods:

X-ray crystallography was explained very poorly in terms of data collection and processing (solving and refining of the model)

Was DFT calculations performed for an isolated structure in vacuum or in periodic or quasiperiodic manner. Is the X-ray structure was firstly optimised before calculations?

Inhibition assays - how the IC50 were calculated (method, software)

Author Response

Response to reviewer 3 comments/suggestions:

Thanks for providing us the opportunity to revise our manuscript entitled “In vitro and In Silico Investigation of Diterpenoid Alkaloids Isolated from Delphinium chitralense ” We agree with the comments and concerns as pointed by your good self. We have addressed them in earnest in this revision. The corrections and suggestions were made in yellow highlight in the revised article and the responses to each comment are addressed below. Please let us know if any other things are necessary for publication. Thank you for all the efforts spent on our manuscript. We are looking forward to your decision on our revised manuscript.

Introduction:
Reviewer Comment: A figure showing the generalized structure of discussed alkaloids should be added, otherwise it is difficult to follow the text, where importance of several functional groups is highlighted.
Authors response: General structures of C-20 and C-19 diterpenoid alkaloids with numbering is provided in figure-1.

Reviewer Comment: line 34 – interpunction

Authors response: Corrected.

Reviewer Comment: lines 36-37 - these details about folk medicine seem to be unnecessary and should be removed.

Authors response: Removed as per suggestion.

Results:
Reviewer Comment: Figure 1 - no numeration of C atoms, it need to be added in accordance with the numeration used for NMR analysis in tables 1-2.

Authors response: In Figure 1, now may be read as figure 2, all Carbons are numbered according to the numbering pattern used for all C20 diterpenoids alkaloids.  
Reviewer Comment: Figure 2 - no numeration of C atoms...

Authors response: In Figure 2, now may be read as figure 3 , carbons are assign numbers accordingly as per reviewer comments.
Reviewer Comment: Figure 3 - very low resolution, it need to be reuploaded in higher quality

Authors response: Quality of figure 3, now may be read Figure 4 is improved as per suggestion.
Reviewer Comment: Table 3 - a significant number of crystallographic parameters: numbers of reflections, independent reflections, data/restrains/parameters or goodness of fit among others is lacking and need to be provided.

Authors response: Table 3, now table 2, is revised as per comments.
Reviewer Comment: Table 6 - bad formatting, numbering of tables 5 and 6 is wrong, need to be inverted

Authors response: Table 6 now table 5 is reformatted and revised accordingly.

Reviewer Comment: Following tables (X-ray crystallography of comp. 1) must be provided:
bond lengths, valenca angles, torsion angles, geometry of hydrogen bonds (if applies).

Authors response: CIF file is provided as supplementary material. Cambridge Crystallographic Data Centre as supplementary publication No. CCDC 1470285. www.ccdc.cam.ac.uk/getstructures
Discussion:

Reviewer Comment: Most of this part need to be moved to "Results" since it is merely description of experimental and computational studies, perhaps both parts can be merged into one - results and discussion. Lack of numbering of carbon atoms on figures makes discussion of NMR spectra very difficult to follow

Authors response: Both results and discussion are merged into single Results and Discussion as per suggestion and has been revised accordingly.

Reviewer Comment: Preparation of compounds 1-3 should be moved to Materials and Methods

Authors response: The isolation procedure of compounds 1-3 has been moved to material and methods section.

Reviewer Comment: Description of crystal structure of compound 1 (including its stereochemistry) is very limited. Is there any particular explanation why stereochemistry of 1 cannot be elucidated from X-ray data?

Authors response: Crystal structure determination of 1 is revised with proper citation regarding absolute configuration.

Reviewer Comment: What was the rationale for DFT calculations? The idea behind this part of the study and the results are almost neither presented nor discussed in the text

Authors response: Rational for DFT is to get additional information about the new natural product regarding their optimized geometry and other electronic properties.

The following was added into results and discussion.

The DFT simulations are necessary to gain the deeper insights into the molecular structure and electronic properties as recently lot of literature reported where the DFT simulations are used along with experimental study to obtain the electronic properties which are harder to obtain through experimental analysis. Therefore, we performed the DFT calculations and studied different properties for the studied compound like HOMO-LUMO analysis, reactivity, global hardness and optimized structure etc. The results are discussed in the results and discussion chapter under 2.1.2 section highlighted in yellow.

Reviewer Comment: On which base the mechanism of inhibition (competitive for compounds 1-3) was inferred. Kinetic plots for enzymatic assays should be also provided here or in the supplementary.

Authors response: On the basis of mechanism-based kinetic study (Lineweaver-Burk, Dixon plots and their replots), competitive type inhibition is observed.

Reviewer Comment: What is the unit of energy obtained for docking? Why there is almost no discussion concerning the mechanism of binding and differences between protein-ligand interactions for each compound.

Authors response: The compounds-receptors docking interaction were calculated in term of binding free energy. The protein-ligand binding free energy calculations based on atomistic molecular simulations promise to play a growing role in drug discovery, as they provide estimates of the binding affinities of compounds proposed as drug candidates for a protein target. Here in current study binding free energy calculated in unit of Kcal-mol-1. Apologies this unit was not mentioned in previous draft. Now it has been revised and mentioned in the new draft.

Our in silico analyses identified that all the three compounds feasibly interact with the active site gorge of acetylcholinesterase and this indicate the competitive inhibitory nature of these compounds. This is more obvious that the difference in protein-ligand interaction among these compounds is due to basic chemical structure differences.  

Materials and methods:

Reviewer Comment: X-ray crystallography was explained very poorly in terms of data collection and processing (solving and refining of the model).

Authors response: Revised as per suggestion

Reviewer Comment: Was DFT calculations performed for an isolated structure in vacuum or in periodic or quasiperiodic manner. Is the X-ray structure was firstly optimised before calculations?

Authors response: The DFT simulations are performed for non-periodic molecular structure. Initially the structure is modeled according to the XRD analysis and then it was optimized using DFT tools and then various electronic properties were calculated for the optimized structure. 

Reviewer Comment: Inhibition assays - how the IC50 were calculated (method, software)

Authors response: The calculation of IC50 values have been provided in material and method section.

Various dilutions of the tested compounds (62.5, 125, 250, 500 and 1000 μg/mL) were used in this assay for IC50 determination. The data (activity in percent vs concentration) was fitted into a non-linear sigmoid plot using MS excel (Microsoft) program. The non-linear concentration-dependent inhibitory concentration of compounds were taken into account at low and high concentration   leading to determination of IC50 values. The effective concentrations of compounds 1-3 were represented into μM.

Reviewer 4 Report

Dear authors

The MS entitled “ In vitro and In silico Investigation of Diterpenoid Alkaloids Isolated From Delphinium chitralense” has thoroughly been reviewed. The article describes isolation and characterization of some new diterpene alkaloids from medicinal plant Delphinium chitralense”. The authors described the procedures in a very well manner while the data seems to be original. Furthermore, the metabolites show strong enzyme inhibitory potential which is well established by supportive evidences. Some of the corrections need to be addressed as per my suggestions. 

1.       The article should be checked for according to the journal format.

2.       Provide email addresses of all the authors.

3.       The key words should be explained before abbreviation.

4.       Amend the title for more reading interest.

Introduction

Line 40: huge should be changes into large.

Line 73. A brief description of ligand-enzyme docking and brief aspect of active cite interaction as per previous studies.

Results

Kindly provide the NMR spectrum (both 1H and 13C) for compound 1 for checking of accuracy.

Line 122. How the compound was crystallized? Did the authors carry some extra experiments for crystallization?

Line 152: Why basified on pH 8-10?

Line 168: please provide 2D HMBC spectrum for compound 3.

Experimental section

Line 221: was only Dragendorff’s reagent used for detection? Why any other reagent not used?

Line 224: kindly provide the location details (longitude, latitude, altitude) of the place.

Line 236: FFC should be described.

Line 238: write the numbers of compounds (1, 2 or 3) in front of their trivial names.

Line 287: kindly add the statistical analysis in method.

Conclusions

Line 301: correct it.

Also, Check the MS for sentence structures, grammar and punctuation errors.

References

Line 330: Correct the references according to the journal format for example reference 2 and 5. Plant Scientific names should be in italic so make the corrections throughout the MS.

 Regards.

Author Response

Response to reviewer 4 comments/suggestions:

Thanks for providing us the opportunity to revise our manuscript entitled “In vitro and In Silico Investigation of Diterpenoid Alkaloids Isolated from Delphinium chitralense ” We agree with the comments and concerns as pointed by your good self. We have addressed them in earnest in this revision. The corrections and suggestions were made in yellow highlight in the revised article and the responses to each comment are addressed below. Please let us know if any other things are necessary for publication. Thank you for all the efforts spent on our manuscript. We are looking forward to your decision on our revised manuscript.

Main Comment:

The MS entitled “ In vitro and In silico Investigation of Diterpenoid Alkaloids Isolated From Delphinium chitralense” has thoroughly been reviewed. The article describes isolation and characterization of some new diterpene alkaloids from medicinal plant Delphinium chitralense”. The authors described the procedures in a very well manner while the data seems to be original. Furthermore, the metabolites show strong enzyme inhibitory potential which is well established by supportive evidences. Some of the corrections need to be addressed as per my suggestions. 

Reviewer Comment: The article should be checked for according to the journal format.

Authors response: The manuscript has been revised according to the standard format of journal guidelines as per reviewer comments.

Reviewer Comment: Provide email addresses of all the authors.

Authors response: Email addresses of all authors have been included as per reviewer suggestion.

Reviewer Comment: The key words should be explained before abbreviation.

Authors response: Corrected as per suggestion.

Reviewer Comment: Amend the title for more reading interest.

Authors response: Current title of the manuscript is much relevant to the special issue of the journal and research caried out.

Introduction

Reviewer Comment: Line 40: huge should be changes into large.

Authors response: Corrected as per suggestion.

Reviewer Comment: Line 73. A brief description of ligand-enzyme docking and brief aspect of active cite interaction as per previous studies.

Authors response: Corrected as per suggestion.

Results

Reviewer Comment: Kindly provide the NMR spectrum (both 1H and 13C) for compound 1 for checking of accuracy.

Authors response: The NMR Spectra of compound 1 are provided below as per suggestion.

Reviewer Comment: Line 122. How the compound was crystallized? Did the authors carry some extra experiments for crystallization?

Authors response: Compound 1 was purified by repeated column chromatography as white crystal. To get the crystal in excellent shape, it was re-crystalized by using mixture of solvents and very slow evaporation.

Reviewer Comment: Line 152: Why basified on pH 8-10?

Authors response: To get maximum yield of alkaloidal fraction, the crude extract is general basified up to pH 8-10.

References: S. Ahmad, H. Ahmad, H.U. Khan, A. Shahzad, E. Khan, S.A. Ali Shah, M. Ali, A. Wadud, M. Ghufran, H. Naz, M. Ahmad, Crystal structure, phytochemical study and enzyme inhibition activity of Ajaconine and Delectinine, JMoSt. 1123 (2016) 441–448. https://doi.org/10.1016/J.MOLSTRUC.2016.06.051.

  1. Ahmad, S. Ahmad, S.A.A. Shah, H.U. Khan, F.A. Khan, M. Ali, A. Latif, F. Shaheen, M. Ahmad, Selective dual cholinesterase inhibitors from Aconitum laeve, Https://Doi.Org/10.1080/10286020.2017.1319820. 20 (2017) 172–181. https://doi.org/10.1080/10286020.2017.1319820.

Reviewer Comment: Line 168: please provide 2D HMBC spectrum for compound 3.

Authors response: The 2D NMR Spectra of compound 3 are provided below as per comments.

Experimental section

Reviewer Comment: Line 221: was only Dragendorff’s reagent used for detection? Why any other reagent not used?

Authors response: Dragendorff’s reagent is generally used for the detection of alkaloid because it is more versatile and can be stored for long time. Beside this we have also used the Iodine vapours, but rarely.

Reviewer Comment: Line 224: kindly provide the location details (longitude, latitude, altitude) of the place.

Authors response: Revised as suggested.

Reviewer Comment: Line 236: FFC should be described.

Authors response: FCC stand for Flash column chromatography

Reviewer Comment: Line 238: write the numbers of compounds (1, 2 or 3) in front of their trivial names.

Authors response: Corrected accordingly

Reviewer Comment: Line 287: kindly add the statistical analysis in method.

Authors response: Revised as per suggestion

Conclusions

Reviewer Comment: Line 301: correct it.

Authors response: Conclusion is rephrased.

Reviewer Comment: Also, Check the MS for sentence structures, grammar and punctuation errors.

Authors response: The manuscript is thoroughly checked for all sort of error as per suggestion.

References

Reviewer Comment: Line 330: Correct the references according to the journal format for example reference 2 and 5. Plant Scientific names should be in italic so make the corrections throughout the MS.

Authors response: Corrected accordingly.

Round 2

Reviewer 2 Report

The authors present a revision of manuscript entitled “In vitro and in silico investigation of diterpenoid alkaloids isolated from Delphinium chitralense”. The manuscript has been revised accordingly, but it still shows a few mistakes such as unit of density (mg m-3). Overall, the manuscript can be accepted now.

Author Response

Response to reviewer 2 comments/ suggestions.

Reviewer 2 comments: The authors present a revision of manuscript entitled “In vitro and in silico investigation of diterpenoid alkaloids isolated from Delphinium chitralense”. The manuscript has been revised accordingly, but it still shows a few mistakes such as unit of density (mg m-3). Overall, the manuscript can be accepted now.

Author response: Correction has been made as per suggestion.